# TIME SERIES CAUSAL DISCOVERY VIA INSTANCE-SPECIFIC MODELING AND INTERVENTION-BASED PRETRAINING

## ABSTRACT

Causal discovery in time series is crucial for downstream tasks such as tracing the root causes of anomalies. Structural Causal Models (SCMs) provide a principled way to formalize the generative processes of observed data. However, learning generalized causal models for time series remains challenging due to the lack of detailed modeling and limited generalization ability. In this paper, we propose **T-Caus**, a novel pretraining framework designed to improve the generalization of **T**ime series **Caus**al models across diverse downstream tasks. To capture complex temporal causal dependencies, T-Caus introduces a hierarchical instance-specific temporal causal discovery framework that employs a dual-scale iterative attention to enhance window-level causal relationships, and a Gaussian mixture with an instance-level routing mechanism to handle heterogeneous exogenous distributions. To further address distribution shifts across time series, T-Caus adopts generalizable causal learning with causal invariance, which explicitly leverages intervention-based learning and a causal mixup strategy to promote stable causal discovery and stronger generalization. Extensive experiments on multiple real-world out-of-distribution (OOD) datasets demonstrate that T-Caus exhibits strong generalization, achieving superior performance in both causal discovery and root cause identification. The code and datasets are available at the link.

## 1 INTRODUCTION

Accurate causal discovery from observed time series is fundamental to many real-world applications. For instance, it can reveal the drivers of stock price fluctuations (Li et al., 2024) or the factors influencing river water levels (Stein et al., 2025), thereby enabling actionable insights and more robust decision-making. In root cause diagnosis (Han et al., 2025; Nagalapatti et al., 2025), constructing a causal graph facilitates the identification of the source of system failure and allows tracing how these failures propagate. Similarly, in anomaly detection (Liu et al., 2025; Kim et al., 2025), monitoring changes in causal relationships can effectively reveal anomalous events.

Recent studies have explored pre-training for causal inference. CInA(Zhang et al., 2024) proposes a foundation model for treatment effect estimation, while Causal Pretraining(Stein et al., 2024) adopts a supervised approach to predict the existence of edges in causal graphs. Leveraging the generalization ability of this paradigm, we investigate pre-training for discovering window causal graphs from time series data. Although pre-training has shown promise for causal graph inference, advances in model architecture and generalization remain necessary to fully realize its potential.

**Challenge 1:** *Inherent complexity of temporal causal dependencies*. As shown in Figure 1, channel 3 at time $t$ is influenced by channel 1 at $t-2$, which in turn is influenced by channel 2 at $t-4$. This reveals an iterative, cascading dependency across time steps. Existing methods (Cheng et al., 2023; Han et al., 2025) suffer from insufficient modeling of such cross-channel interactions and cascading temporal dependencies. It is crucial to capture causal relationships not only within individual windows but also across multiple time windows. Furthermore, in a structural causal model, observed time series (endogenous variables) are influenced not only by other endogenous variables but also by unobserved factors (exogenous variables). These exogenous variables, as external factors independent of the system's internal structure, provide critical signals for root cause identification (Han

et al., 2025). Accurately modeling their influences is thus essential for reliable discovery of causal graphs. However, the distributions of exogenous variables are diverse across different time series. Existing approaches (Han et al., 2025), which typically assume a fixed Gaussian prior, lack the flexibility to adapt to such diversity and cannot model exogenous factors across multiple series.

**Challenge 2:** *Causality-Aware Generalization under Distribution Shifts.* In real-world scenarios, continuously generated data streams often exhibit non-stationarity, leading to distributional shifts over time. Such shifts break the alignment between historical data and future out-of-distribution (OOD) data. Pretrained models are generally expected to handle OOD cases under the Vicinal Risk Minimization principle (Chapelle et al., 2000), which emphasizes modeling the neighborhood of each training example to achieve robustness against unseen variations. However, existing time series causal discovery methods fail to account for both the diver-

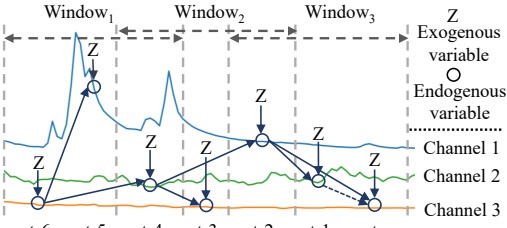

Figure 1: Intra- and inter-window causal dependencies in multivariate time series $x$. Solid lines indicate true causal dependencies and dashed lines represent spurious correlations.

sity of individual series and the variability of causal relationships across different series. Additionally, in causal discovery, distinguishing between correlation and causation is crucial (Pearl & Mackenzie, 2018). As shown in Figure 1, both channel 3 at time $t$ ($x_{3,t}$) and channel 2 at $t-1$ ($x_{2,t-1}$) are influenced by channel 1 at $t-2$, which induces correlation between them. However, this does not imply a direct causal link from $x_{2,t-1}$ and $x_{3,t}$. What truly matters is determining whether a causal relationship exists. Unfortunately, existing deep learning methods lack effective constraints to identify and differentiate spurious correlation from causation.

For **Challenge 1,** we propose a *hierarchical instance-specific temporal causal discovery framework* that models window-level features at multiple scales and captures the instance-level distributions of exogenous variables. Our framework features two key components: (1) *Dual-scale iterative representation enhancement*, which employs an attention-based mechanism to iteratively model both intra-window and inter-window dependencies, capturing fine-grained causal dynamics within windows while preserving coarse-grained long-range causal propagation across windows; and (2) *Instance-specific exogenous variable estimation*, which employs an adaptive mixture-of-Gaussians model with a routing mechanism to enable instance-specific approximation of exogenous variable distributions, thereby enhancing the reliability of causal discovery across diverse time series.

For **Challenge 2,** we propose *generalizable causal learning with causal invariance* to enhance the model's ability to perceive causal relationships and improve its OOD robustness. *Causal pretext task* introduces carefully designed interventions within the time series to eliminate spurious correlations, thereby mitigating their influence on causal structure identification. By treating the prediction of the intervened window as a pretext task, the model is encouraged to exploit discrepancies across varying time series and learn which conditional distributions remain invariant. This invariance serves as a strong signal for identifying the underlying causal relationships, even under distributional shifts or hidden confounders. Furthermore, *Time series causal mixup* mixes both raw time series and their associated causal graphs, generating augmented sequences that exhibit causal dependencies with varying strengths and patterns. This not only enriches the training data with diverse causal scenarios but also forces the model to generalize beyond seen correlation structures.

Specifically, we make the following contributions:

- We propose a hierarchical instance-specific temporal causal discovery framework that captures both inter-window and intra-window dependencies in individual time series, while flexibly handling instance-specific exogenous noises across multiple series.

- We enhance generalization through a pretraining strategy that employs intervention-based pretext tasks to differentiate causation from correlation and reveal causal invariance in time series, alongside causal mixup to achieve smoother gradients and improved robustness.

- Extensive experiments on multiple datasets, covering both causal discovery and root cause identification, demonstrate the superior generalization ability of our method. Experimental results show that T-Caus consistently outperforms existing methods in both tasks.

## 2 RELATED WORK

**Causal Discovery for Time Series.** Granger causality-based methods assume that if past values of $X$ improve the prediction of future values of $Y$, then $X$ is a Granger cause of $Y$. This idea has been extended through various neural architectures. For example, cLSTM (Tank et al., 2021) leverages RNNs to infer Granger causal structures; TCDF (Nauta et al., 2019) utilizes attention-based CNNs for efficient and interpretable causal discovery; GVAR (Marcinkevics & Vogt, 2021) adopts a

Table 1: Comparison of causal discovery methods. The symbol ✓ indicates that a component is used in the model, × indicates that it is not used.

| Methods | Fine-grained | Coarse-grained | Exogenous | Generalization |
|---|---|---|---|---|
| CDMI (Ahmad et al., 2024) | ✓ | × | × | × |
| CUTS (Cheng et al., 2023) | ✓ | × | × | × |
| TCDF (Nauta et al., 2019) | ✓ | ✓ | × | × |
| CP (Stein et al., 2024) | × | ✓ | × | ✓ |
| AERCA (Han et al., 2025) | ✓ | × | ✓ | × |
| T-Caus | ✓ | ✓ | ✓ | ✓ |

vector autoregressive model with generalized coefficient matrices to increase modeling flexibility; and CUTS (Cheng et al., 2023; 2024) constructs a causal adjacency matrix directly from the data under sparsity regularization. However, Granger causality-based approaches share a fundamental limitation: they do not explicitly model endogenous errors or exogenous noise. As a result, their applicability in practical scenarios can be restricted, especially in root cause diagnosis tasks where measurement inaccuracies and stochastic disturbances are common. Structural Causal Model (SCM) approaches explicitly characterize the functional relationships among endogenous variables while modeling the influence of exogenous noise. Varlingam (Hyvärinen et al., 2010) represents a restricted form of structural equation modeling (SEM), combining a non-Gaussian instantaneous causal model with vector autoregressive dynamics. TiMINo (Peters et al., 2013) makes a stronger assumption that exogenous noise variables are independent over time. AERCA (Han et al., 2025) leverages an autoencoder to simulate the data generation process, explicitly modeling both the causal relationships and the distribution of exogenous variables. As summarized in Table 1, existing methods largely overlook distribution shifts and struggle to jointly model endogenous variables and exogenous noise across multiple scales. To address these gaps, we propose a time series causal discovery framework via instance-specific modeling and the intervention-based pretraining.

**Pre-trained Causal Discovery.** Recent efforts have integrated causal inference into pre-trained models. CaML (Nilforoshan et al., 2023) formulates personalized effect prediction as a meta-task for zero-shot generalization; Cond-FiP (Mahajan et al., 2024) dynamically infers structural causal models via a Fixed-Point Approach (Scetbon et al., 2024); and CInA (Zhang et al., 2024) learns transferable causal representations from unlabeled data, enabling zero-shot inference without fine-tuning. However, these methods do not explicitly capture temporal causality. CP (Stein et al., 2024) tackles time series by learning window-based causal graphs through supervised training across four architectures, yet it overlooks key temporal properties such as multi-scale dependencies, exogenous variable effects, distribution shifts, and evolving causal relations (see Table 1). To overcome these limitations, we propose a pre-training framework for temporal causal discovery that (i) captures hierarchical dynamics through dual-scale interactions at the window level and instance-specific temporal patterns at the instance level, and (ii) enhances generalizable causal learning by leveraging causal mixup and intervention to encode causal invariance. This framework improves OOD performance and enhances causal perception in dynamic environments.

## 3 PROBLEM FORMULATION

In this work, we consider a multivariate time series $\boldsymbol{x} = \{x_{1:T,i}\}_{i=1}^C$, where each $x_{1:T,i}$ represents a sequence of $T$ observations on the $i$-th channel, and $C$ denotes the number of channels. Following Han et al. (2025), we assume that the data are generated by a time-invariant structure causal model (SCM) of the form:

$$x_{t,i} = \sum_{k=0}^{n-1} \sum_{j=1}^C G_{t-n+k,j,i} \cdot f_{k,j,i}(x_{t-n+k,j}) + Z_{i,t}, \tag{1}$$

where $n$ is the maximum time lag, $\boldsymbol{G} \in \mathbb{R}^{n \times C \times C}$ is the causal graph, and $G_{t-n+k,j,i}$ indicates the impact of channel $j$ at time step $t-n+k$ on $x_{t,i}$, $f_{k,j,i}(\cdot)$ is a function that represents a nonlinear transformation of the past observations, and $Z_{i,t}$ is an exogenous term for channel $i$ at time step $t$.

The goal of causal discovery is to learn a model $g_\theta(\cdot)$ that maps the observed time series $x$ to the corresponding window causal graph $G = g_\theta(x)$ (Assaad et al., 2023; Han et al., 2025).

# 4 METHODOLOGY

We propose **T-Caus**, a generalizable time series causal discovery framework that leverages instance-specific modeling and interventions-based pretraining. The framework is shown in Figure 2. To capture causal dependencies and handle exogenous variables, we introduce a **hierachical instance-specific causal discovery framework**. The dual-scale iterative representation enhancement module uncovers causal relationships through an alternating, attention-based fusion of intra-window and inter-window features, capturing fine-grained local causal dynamics within windows while preserving long-range causal dependencies across windows. The instance-specific exogenous variable estimation module utilizes an adaptive mixture-of-Gaussians model with a routing mechanism, enabling tailored approximation of exogenous variable distributions for each instance. This enhances the robustness of causal discovery across diverse and heterogeneous time series. To enhance the generalizability of causal discovery, we introduce **generalizable causal learning with causal invariance strategy**, which consists of two components. The intervention pretext task performs interventions on the generative process of time series and formulates a prediction task for the intervened window, thereby eliminating spurious correlations and enabling stable causal discovery across different environments. Causal Mixup enhances the stability of the model's causal dependencies by mixing both time series data and their corresponding causal graphs. Finally, the observed time series is reconstructed by jointly leveraging the learned causal structure, endogenous dynamics, and inferred exogenous factors, ensuring a comprehensive and interpretable modeling of temporal data.

## 4.1 HIERARCHICAL INSTANCE-SPECIFIC TEMPORAL CAUSAL DISCOVERY FRAMEWORK

**Dual-scale Iterative Representation Enhancement.** In time series modeling, sequences are often partitioned into sliding windows. Given a multivariate time series $x \in \mathbb{R}^{T \times C}$, we divide it into $m$ windows of size $(n + 1)$, denoted as $s \in \mathbb{R}^{m \times (n+1)C}$. Each window consists of past observations $s_{lag} \in \mathbb{R}^{m \times nC}$, and the current step $s_t \in \mathbb{R}^{m \times C}$. Intra-window features capture local dynamics, and inter-window relations provide broader context. To jointly model these complementary scales, we propose an alternating attention mechanism that iteratively integrates features within and across windows. First, we apply self-attention within each window. The sequence $s$ is projected into $Q_{intra}^s$, $K_{intra}^s$, and $V_{intra}^s$, and the intra-window representation is then computed as:

$$\tilde{s}_{intra} = \text{Softmax}\left( Q_{intra}^s (K_{intra}^s)^\top / \sqrt{d_k} \right) V_{intra}^s \tag{2}$$

Next, we perform inter-window self-attention. The intra-window outputs $\tilde{s}_{intra} \in \mathbb{R}^{m \times (n+1)C}$ are reshaped into $s_{inter} \in \mathbb{R}^{(n+1)C \times m}$, and projected into $Q_{inter}^s$, $K_{inter}^s$, and $V_{inter}^s$. The inter-window representation is then computed as:

$$\tilde{s}_{inter} = \text{Softmax}\left( Q_{inter}^s (K_{inter}^s)^\top / \sqrt{d_k} \right) V_{inter}^s \tag{3}$$

The output $\tilde{s}_{inter}$ is reshaped back to the original space and fed into the next iteration. Repeating this alternating process for $N$ iterations progressively refine the representation, yielding the dual-scale enhanced representation $\tilde{s} \in \mathbb{R}^{m \times (n+1)C}$. Similar to $s$, $\tilde{s}$ is partitioned into $\tilde{s}_{lag}$ for past observations and $\tilde{s}_t$ for the current step. We then encode $\tilde{s}_{lag}$ and $\tilde{s}_t$ separately, and compute their inner product to obtain the probability matrix $\tilde{G}_p$ and the weight matrix $\tilde{G}_a$:

$$u_p = \text{MLP}(\tilde{s}_{lag}), \quad v_p = \text{MLP}(\tilde{s}_t), \quad \tilde{G}_p = u_p^T v_p, \quad u_p \in \mathbb{R}^{m \times nC}, \quad v_p \in \mathbb{R}^{m \times C} \tag{4}$$

$$u_a = \text{MLP}(\tilde{s}_{lag}), \quad v_a = \text{MLP}(\tilde{s}_t), \quad \tilde{G}_a = u_a^T v_a, \quad u_a \in \mathbb{R}^{m \times nC}, \quad v_a \in \mathbb{R}^{m \times C}$$

Here, $\tilde{G}_p$ is a binary matrix indicating the presence or absence of causal links, while $\tilde{G}_a$ quantifies the corresponding edge weights. Finally, the causal graph is trained to align the estimated link probability matrix $\tilde{G}_p$ and the edge weight matrix $\tilde{G}_a$ with their ground-truth matrices $G_p$ and $G_a$:

$$\mathcal{L}_G = -(G_p \cdot \log(\tilde{G}_p) + (1 - G_p) \cdot \log(1 - \tilde{G}_p)) + \| \tilde{G}_a - G_a \|_2 . \tag{5}$$

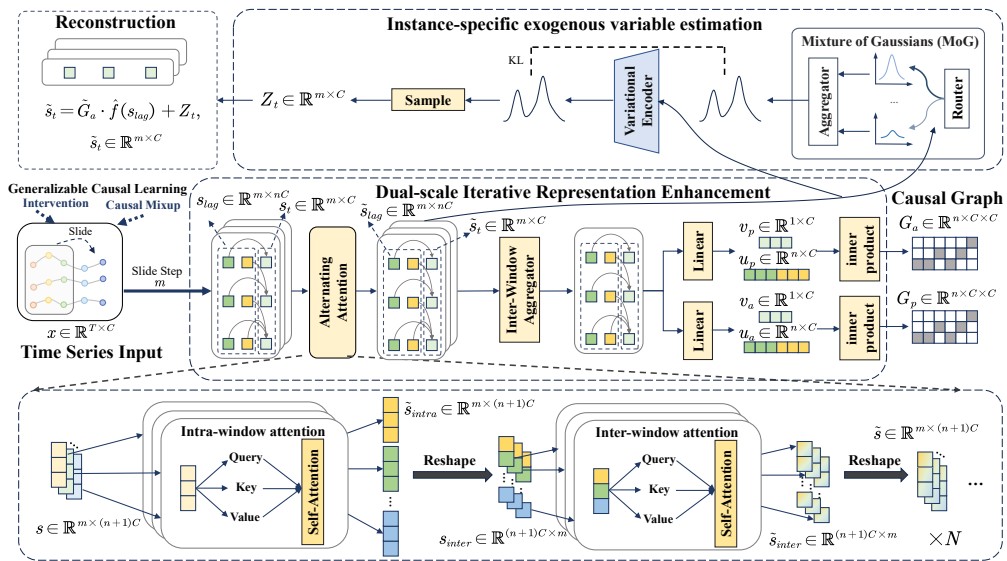

Figure 2: The framework of hierarchical instance-specific temporal causal discovery consists of two main modules: (1) Dual-scale Iterative Representation Enhancement, (2) Instance-specific exogenous variable estimation. Furthermore, we enhance the model's generalization ability through generalizable causal learning with causal invariance, leveraging intervention-based learning and causal mixup strategies.

**Instance-specific exogenous variable estimation.** According to Equation 1, we define a data-generating distribution $p(D)$, which encapsulates both the underlying causal structure and the influence of exogenous factors. The observed time $x \sim p(D)$ is generated by first sampling from the causal structure distribution $p(G)$ and the prior distribution of exogenous variable $p(z)$, and then iteratively generating data through the data-generating mechanism $p(D|G, z)$:

$$x \sim p(D) = \int_G \int_z p(D \mid G, z) \cdot p(G) \cdot p(z) dz dG. \qquad (6)$$

In multivariate time series, noise and external disturbances often exhibit diverse and dynamic patterns, reflecting the heterogeneity of exogenous influences. This observation suggests that a single, globally shared Gaussian distribution is insufficient to capture the variability of exogenous factors. To address this, we assume a prior distribution over the exogenous variables in the form of a mixture of Gaussians. To further capture instance-specific variability, we adopt an adaptive Gaussian mixture model, where the mixing coefficients $\boldsymbol{\pi} \in \mathbb{R}^K$ are computed from enhanced time representation $\tilde{s}$ via a routing mechanism. The means $\mu = 0$ and variances $\Sigma$ of the $K$ Gaussian component is treated as learnable parameters:

$$\boldsymbol{\pi} = \text{Router}(\tilde{s}), \quad p(z) = \sum_{k=1}^{K} \pi_k \mathcal{N}(z \mid \mu_k, \Sigma_k), \quad \sum_{k=1}^{K} \pi_k = 1. \qquad (7)$$

We employ a variational encoder to approximate the posterior distribution $q(z)$ of the exogenous variables. Using the reparameterization trick, we sample the exogenous variable $Z_t$ from $q(z)$ at each time step $t$. The sampled variable $Z_t$ is then incorporated into the reconstruction of the current time representation $\tilde{s}_t$, together with the predicted causal graph $\tilde{G}_a$, and the lagged endogenous variables $s_{lag}$. In this manner, both exogenous influences and the causal dependencies are explicitly modeled, ensuring that the learned representations are consistent with a structural causal model. The reconstruction of $\tilde{s}_t$ is formulated as:

$$\tilde{s}_t = \tilde{G}_a \cdot \tilde{f}(s_{lag}) + Z_t, \qquad (8)$$

where $\tilde{f}$ denotes a multi-layer perceptron.

To regularize the latent space, we further introduce a KL divergence term that measures the discrepancy between the learned posterior distribution $q(z)$ and the assumed prior mixture Gaussian

distribution $p(z)$:

$$\mathcal{L}_{KL}(p(z) \parallel q(z)) = \int p(\mathbf{z}) \log \frac{p(\mathbf{z})}{q(\mathbf{z})} \, d\mathbf{z}, \tag{9}$$

## 4.2 Generalizable Causal Learning with Causal Invariance

**Intervention Pretext Task.** Time series data often exhibit autocorrelation and interdependencies, where past values influence the present, and multivariate series may involve cross-channel causality. Relying solely on observational data can lead to confusion between correlation and causality. Intervening on a variable is an effective way to alter its data-generating mechanism. For example, $do(X = x)$ intervenes by setting $X$ to $x$ exogenously, cutting off all causal influences from its parents in the causal graph, which can be proved in Theorem 1.

**Theorem 1.** *Let $X_{1,t-2}$ be a potential confounder of the causal relationship $X_{2,t-1} \to X_{3,t}$ in SCM. In the observational distribution, suppose $X_{1,t-2}$ and $X_{2,t-1}$ are statistically dependent. The observational distribution induces a* spurious correlation *between $X_{2,t-1}$ and $X_{3,t}$ through the backdoor path $X_{2,t-1} \leftarrow X_{1,t-2} \to X_{3,t}$. This creates a biased estimate of the true causal effect. In contrast, the interventional distribution blocks this path by replacing the conditional $P(X_{1,t-2} \mid X_{2,t-1})$ with the marginal $P(X_{1,t-2})$, thereby eliminating the influence of spurious associations and recovering the genuine causal effect of $X_{2,t-1}$ on $X_{3,t}$. For detailed proofs, see Appendix A.1.*

We propose a method that leverages designed interventions to learn causal structure. Specifically, we begin by randomly selecting a time interval $[t_1, t_2]$. Next, we randomly sample a value from a lag window preceding $t_1$ and replace it with the value at $t_1$. Based on the underlying causal dynamics, we iteratively compute the intervened values $(x'_{t_1}, \cdots, x'_{t_2})$, which replace the original segment $(x_{t_1}, \cdots, x_{t_2})$. This yields a modified time series $x^{do}$ containing both intervened and non-intervened windows. By examining segmented windows, we can then identify which windows are inconsistent with the causal structure to obtain the ground-truth of binary intervention labels.

To enhance generalization and encourage the model to capture stable, invariant causal dependencies, we train it to predict the window in which an intervention occurs. At the same time, to preserve semantic consistency under interventions, we apply contrastive learning so that representations of the intervened series $x^{do}$ remain close to those of the original $x$. Specifically, we obtain intervention-based representations $u^{do}$ and $v^{do}$ and align them with the original $u$ and $v$ using Eq. 4 through contrastive loss, while also incorporating a classification loss for intervention detection:

$$\mathcal{L}_{do} = -\sum_{i=1}^{N_u} \log \frac{\exp\big(\text{sim}(u_i, u^{do,+})/\tau\big)}{\displaystyle\sum_{j=1}^{N_{u^{do}}} \exp\big(\text{sim}(u_i, u_j^{do})/\tau\big)} - \sum_{i=1}^{N_v} \log \frac{\exp\big(\text{sim}(v_i, v^{do,+})/\tau\big)}{\displaystyle\sum_{j=1}^{N_{v^{do}}} \exp\big(\text{sim}(v_i, v_j^{do})/\tau\big)} \tag{10}$$
$$+ \mathbb{E}_{(c,\hat{c})} \left[ -c \log(\hat{c}) - (1-c) \log(1-\hat{c}) \right],$$

where $c$ and $\hat{c}$ denote the ground truth and predicted intervention labels, $N_u$ and $N_v$ are the numbers of $u$ and $v$, $N_{u^{do}}$ and $N_{v^{do}}$ are their intervened counterparts. $\tau$ is a temperature parameter, $sim(\cdot, \cdot)$ is the similarity function, and $u^{do,+}$ denotes the positive pair of $u_i$ from the same time series.

**Time Series Causal Mixup.** Inspired by the *mixup* (Zhang et al., 2018), we propose a time series causal mixup strategy for causal discovery in temporal data. This operation smooths the decision boundary, leading to more stable and continuous estimates of edge existence and causal strength. Specifically, we randomly sample $k$ time series instances $x^i$ and their corresponding causal relations $G_a^i$ from the training dataset to construct mixed combinations, formalized as:

$$x^m = \sum_{i=1}^{k} \lambda_i x^i, \quad G_a^m = \sum_{i=1}^{k} \lambda_i G_a^i, \tag{11}$$

where the mixing coefficient $\lambda_i$ is drawn from a symmetric Dirichlet distribution (Ansari et al., 2024). By blending samples from diverse time series and integrating causal effects of varying strengths, Time Series Causal Mixup achieves smoother gradient flow during training, enhancing the robustness of causal dependency modeling. To learn causal invariance, we minimize the reconstruction loss over three types of time series: $\mathcal{L}_{recon} = \|\tilde{s}_t - s_t\|_2 + \|\tilde{s}_t^m - s_t^m\|_2 + \|\tilde{s}_t^{do} - s_t^{do}\|$, where $s_t$, $s_t^m$, and $s_t^{do}$ are obtained by applying a sliding window to $x$, $x^m$, and $x^{do}$, respectively, and $\tilde{s}_t$, $\tilde{s}_t^m$, and $\tilde{s}_t^{do}$ are computed from $x$, $x^m$, and $x^{do}$ using Eq. 8.

### 4.3 Causal Learning and Root Cause Identification

**Pretraining Phase.** During pretraining, we generate synthetic datasets (see Appendix A.2.1) to train the model. Given a time series $x$, the objective function consists of several components: the reconstruction loss $\mathcal{L}_{recon}$, the causal graph loss $\mathcal{L}_G$, the KL divergence loss $L_{KL}$, the intervened loss $\mathcal{L}_{do}$. The overall pre-training loss is defined as:

$$\mathcal{L}_{pre} = \mathcal{L}_{recon} + \lambda_G \mathcal{L}_G + \lambda_{KL} \mathcal{L}_{KL} + \lambda_{do} \mathcal{L}_{do}, \tag{12}$$

where $\lambda_G$, $\lambda_{KL}$, $\lambda_{do}$ are the corresponding weighting coefficients.

**Fine-tuning Phase.** During fine-tuning, we address two types of downstream tasks. For *Causal Discovery*, where the true data generation process is typically unavailable in real-world scenarios,

we focus solely on the reconstruction loss with $s_t$, the causal graph loss with the link probability matrix $G_p$, and the KL divergence loss $L_{KL}$. The fine-tuning loss is as follows:

$$\mathcal{L}_{ft} = \|\tilde{s}_t - s_t\|_2 - \lambda_G(G_p \cdot \log(\tilde{G}_p) + (1 - G_p) \cdot \log(1 - \tilde{G}_p)) + \lambda_{KL} \mathcal{L}_{KL}. \tag{13}$$

For *Root Cause Identification*, Following AERCA (Han et al., 2025), we perform fine-tuning on normal data to learn the mean ($\mu$) and standard deviation ($\sigma$) of the exogenous variable distribution:

$$\mathcal{L}_{rc} = \|\tilde{s}_t - s_t\|_2 + \lambda_{KL} \mathcal{L}_{KL}. \tag{14}$$

The root cause score is then computed as the z-score $score_t = \frac{Z_t - \mu}{\sigma}$ and Streaming Peaks-Over-Threshold (SPOT) (Siffer et al., 2017) is used to adaptively determine the detection threshold.

## 5 Experiments

### 5.1 Experimental Design

**Datasets.** We construct a synthetic time series causal dataset (see Appendix A.2.1) to pre-train our model, and then fine-tune it on two real-world tasks to adapt to practical scenarios. *Causal Discovery*. We evaluate on a benchmark derived from German river systems (Stein et al., 2025), which covers causal graphs in Eastern Germany (666 stations) and Bavaria (494 stations). The benchmark provides datasets featuring two types of variable quantities and five causal relationship types, including Close, Root Cause, Random+1, Confounder, and Random. Our model is fine-tuned on the Bavaria dataset and evaluated on the Eastern Germany river datasets. *Root Cause Identification*. We evaluate on two benchmarks. The SWaT dataset (Mathur & Tippenhauer, 2016), collected from a scaled-down water treatment testbed under both normal and cyber-attack conditions, making it valuable for assessing intrusion detection in industrial control systems. The MSDS (Multi-Source Distributed System) dataset (Nedelkoski et al., 2020), generated on an OpenStack-based distributed infrastructure, injects controlled faults to emulate anomalies in multi-source cloud environments. Dataset statistics are summarized in Table 5.

**Evaluation Metrics.** *Causal Discovery*. To avoid the complexity of selecting individualized thresholds, we report the AUROC score under the best-performing hyperparameter configuration as the final evaluation metric. *Root Cause Identification*. Following prior work (Ikram et al., 2022; Li et al., 2022; Yu et al., 2021; Ma et al., 2020; Han et al., 2025), we evaluate using recall at top-$k$, denoted as $Recall@K$. This metric measures the probability of correctly identifying root causes within the top-$k$ highest root cause scores. The details are in Appendix A.2.2.

**Baselines.** *Causal Discovery*. We evaluate representative methods spanning classical and modern paradigms. Classical approaches include PCMCI (Runge et al., 2019), Varlingam (Hyvärinen et al., 2010), Dynotears (Pamfil et al., 2020), VAR (Assaad et al., 2023), CDMI (Ahmad et al., 2024). Among recent pretraining methods, we compare with Causal Pretraining (CP) (Stein et al., 2024), which is implemented through both Transformer-based and GRU-based architectures. *Root Cause Identification*. We compare T-Caus with four baselines: 1) $\varepsilon$-Diagnosis (Shan et al., 2019), which locates root causes via pairwise significance tests between normal and abnormal periods; 2) RCD (Ikram et al., 2022), which learns a partial causal graph and treats intervention targets as root causes; 3) CIRCA (Li et al., 2022), which leverages domain knowledge to build structural causal graphs and identifies nodes with significant parent-child distribution shifts; and 4) AERCA (Han et al., 2025), which models causal dependencies and exogenous variables with an autoencoder and attributes anomalies to perturbed exogenous factors.

Table 2: Causal discovery results of AUROC on Eastern Germany river datasets. The best results are in bold, and the second results are underlined.

| Model / Dataset | VAR | Varlingam | Dynotears | PCMCI | CDMI | CP(Gru) | CP(Transformer) | T-Caus |
|---|---|---|---|---|---|---|---|---|
| Close (3) | 0.81 | 0.79 | 0.50 | 0.64 | 0.81 | 0.79 | 0.75 | **0.84** |
| Close (5) | 0.81 | 0.77 | 0.50 | 0.62 | 0.81 | 0.81 | 0.83 | **0.85** |
| Root cause (3) | 0.79 | 0.77 | 0.56 | 0.70 | 0.75 | 0.78 | 0.84 | **0.88** |
| Root cause (5) | 0.75 | 0.77 | 0.56 | 0.74 | 0.65 | 0.81 | 0.84 | **0.86** |
| Random+1 (3) | 0.80 | 0.84 | 0.52 | 0.83 | 0.82 | 0.82 | 0.82 | **0.87** |
| Random+1 (5) | 0.79 | 0.79 | 0.61 | 0.74 | 0.80 | 0.84 | **0.85** | **0.85** |
| Confounder (3) | 0.71 | 0.68 | 0.53 | 0.66 | 0.63 | 0.64 | 0.65 | **0.81** |
| Confounder (5) | 0.72 | 0.70 | 0.53 | 0.64 | 0.71 | 0.71 | 0.71 | **0.79** |
| Random (3) | 0.82 | 0.79 | 0.50 | 0.65 | 0.80 | 0.77 | 0.81 | **0.85** |
| Random (5) | 0.80 | 0.75 | 0.51 | 0.65 | 0.78 | 0.83 | **0.86** | **0.86** |
| **avg** | 0.78 | 0.77 | 0.53 | 0.69 | 0.76 | 0.78 | 0.80 | **0.85** |

Table 3: Results of root cause identification. The best results are in bold.

| Dataset | Model | Recall@1 | Recall@3 | Recall@5 | Recall@10 | Avg@10 |
|---|---|---|---|---|---|---|
| MSDS | $\epsilon$-Diagnosis | 0.004±0.004 | 0.266±0.002 | 0.452±0.009 | **1.000±0.000** | 0.492±0.001 |
| | RCD | 0.412±0.048 | 0.573±0.010 | 0.984±0.001 | **1.000±0.000** | 0.821±0.012 |
| | CIRCA | 0.454±0.238 | 0.860±0.140 | 0.917±0.084 | **1.000±0.000** | 0.809±0.035 |
| | AERCA | 0.381±0.408 | 0.908±0.062 | 0.974±0.027 | **1.000±0.000** | 0.896±0.037 |
| | T-Caus | **0.515±0.252** | **0.993±0.004** | **0.996±0.003** | **1.000±0.000** | **0.929±0.013** |
| SWaT | $\epsilon$-Diagnosis | 0.075±0.179 | 0.125±0.217 | 0.125±0.217 | 0.375±0.383 | 0.180±0.194 |
| | RCD | 0.000±0.000 | 0.000±0.000 | 0.000±0.000 | 0.300±0.458 | 0.100±0.161 |
| | CIRCA | 0.000±0.000 | 0.000±0.000 | 0.000±0.000 | 0.300±0.458 | 0.100±0.161 |
| | AERCA | 0.220±0.111 | 0.290±0.088 | 0.330±0.048 | 0.455±0.044 | 0.342±0.052 |
| | T-Caus | **0.300±0.132** | **0.450±0.107** | **0.450±0.137** | **0.475±0.141** | **0.440±0.105** |

## 5.2 Experimental Results

*Causal Discovery.* Table 2 presents the results for time series causal graph discovery, demonstrating that T-Caus consistently outperforms all baseline methods across various types of causal relationships. This highlights the effectiveness of its dual-scale iterative representation enhancement and instance-specific exogenous variables estimation in capturing complex causal dependencies. A key advantage of T-Caus lies in its ability to learn environment-invariant causal structures through intervention-based modeling and causal mixup. This approach enables the model to accurately identify true causal dependencies among time series while filtering out spurious correlations caused by confounders or environmental biases. For example, we observe that our model achieves strong causal discovery performance on the Eastern Germany river dataset, despite not being directly trained on it. Notably, compared to CP, our method demonstrates a significant improvement, with an average gain of 5%. These results underscore that T-Caus delivers more robust, reliable, and generalizable causal discovery, particularly in real-world settings subject to distribution shifts.

*Root Cause Identification.* Table 3 shows that T-Caus achieves the best performance on real-world datasets in the root cause diagnosis task. Unlike AERCA, which primarily focuses on intra-window modeling, T-Caus leverages a dual-scale iterative representation enhancement to capture both coarse-grained and fine-grained temporal patterns. This approach enhances the model's ability to detect subtle variations over time, leading to superior performance in identifying the root cause of anomalies. Furthermore, the instance-specific exogenous variable estimation boosts T-Caus to adapt to the complex and often noisy conditions found in real-world data. According to the AC@1 metric, T-Caus excels in accurately identifying the time series with the highest root cause score, demonstrating that incorporating intervention-based pre-training enables more stable detection of anomalous root causes even in the presence of dynamic shifts and external disturbances.

**Ablation Studies.** To assess the impact of different components in T-Caus, we conduct an ablation study on inter-window attention, intra-window attention, mixture of Gaussians exogenous valuables, causal mixup, and intervention. As shown in Table 4, each module contributes to the overall per-

Table 4: Ablation study of causal discovery using the Eastern Germany river dataset. w/o inter, w/o intra, w/o exogenous, w/o causal mixup, w/o intervention represent removing the inter-window attention, intra-window attention, exogenous, causal mixup, and intervention, respectively.

| Model / Dataset | w/o inter | w/o intra | w/o exogenous | w/o causal mixup | w/o intervention | T-Caus |
|---|---|---|---|---|---|---|
| Close (3) | 0.83 | 0.82 | 0.80 | 0.83 | 0.80 | **0.84** |
| Close (5) | 0.84 | 0.82 | 0.81 | 0.83 | 0.81 | **0.85** |
| Root cause (3) | 0.86 | 0.84 | 0.82 | 0.86 | 0.83 | **0.88** |
| Root cause (5) | 0.85 | 0.84 | 0.82 | 0.85 | 0.83 | **0.86** |
| Random+1 (3) | 0.85 | 0.84 | 0.83 | 0.86 | 0.84 | **0.87** |
| Random+1 (5) | 0.83 | 0.83 | 0.82 | 0.83 | 0.81 | **0.85** |
| Confounder (3) | 0.79 | 0.79 | 0.78 | 0.80 | 0.75 | **0.81** |
| Confounder (5) | 0.78 | 0.76 | 0.75 | 0.77 | 0.74 | **0.79** |
| Random (3) | 0.84 | 0.84 | 0.81 | 0.83 | 0.82 | **0.85** |
| Random (5) | 0.85 | 0.84 | 0.83 | 0.84 | 0.83 | **0.86** |
| **avg** | 0.83 | 0.82 | 0.81 | 0.83 | 0.81 | **0.85** |

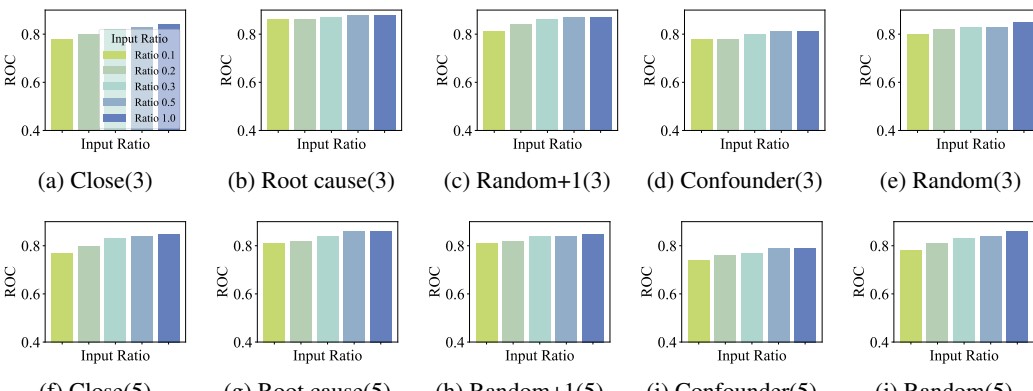

(a) Close(3)  (b) Root cause(3)  (c) Random+1(3)  (d) Confounder(3)  (e) Random(3)

(f) Close(5)  (g) Root cause(5)  (h) Random+1(5)  (i) Confounder(5)  (j) Random(5)

Figure 3: Results under different test input ratios (10%, 20%, 30%, 50%, 100%) on Eastern Germany river dataset (average total sequence length: 3,143).

formance. Notably, removing exogenous variable estimation significantly reduces performance, highlighting the importance of learning instance-specific distributions. Furthermore, the inclusion of intervention yields notable performance improvements, particularly in datasets where the root cause is confounded (Confounder (3) and Confounder (5)). This demonstrates that interventions help the model distinguish between spurious correlations and true causal relationships, enhancing the accuracy of causal graph prediction. Further ablation studies are in Appendix A.2.4.

**Different Time Series Ratio.** Figure 3 shows the impact of varying lengths of the input time series on model performance. We find that using only 30% of the full sequence length yields results comparable to the complete input, while even 10% still provides reasonable results across all datasets. This demonstrates that our method remains effective under limited computational resources and efficient for causal discovery using reduced temporal inputs, making it suitable for real-world applications.

## 6 CONCLUSIONS

This paper introduces T-Caus, a novel pretraining framework designed to improve the generalization of time series causal discovery across diverse downstream tasks. To capture complex temporal dependencies, T-Caus employs a dual-scale attention mechanism for both fine-grained local dynamics and long-range causal relationships, complemented by an instance-specific Gaussian mixture to accommodate heterogeneous exogenous variable distributions flexibly. To further mitigate distribution shifts, T-Caus incorporates a causal mixup strategy and intervention-based learning to encode causal invariance, thereby promoting stable causal discovery and stronger generalization. Experiments on real-world OOD datasets show that T-Caus outperforms existing methods in both causal discovery and root cause identification, establishing it as a robust foundation for time series analysis.

ETHICS STATEMENT

This study is based exclusively on publicly accessible benchmark datasets, as detailed in the paper, with no collection or disclosure of personal or sensitive information. Additionally, no human participants were involved, ensuring full adherence to ethical principles and research integrity standards.

REPRODUCIBILITY STATEMENT

The experimental results and datasets presented in this work are real, and all findings are fully reproducible as described in the paper. Comprehensive details regarding the model architecture, training process, and evaluation methodology are provided in the main manuscript and supplementary materials. To support transparency and enable replication of our work, we have made the source code and datasets publicly available at https://anonymous.4open.science/r/T-Caus-B0CD.

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

# A APPENDIX

## A.1 PROOF OF THEOREM 1

Consider the following linear structural causal model (SCM) with time series channels $X_{1,t-2}, X_{2,t-1}, X_{3,t}$:

$$X_{1,t-2} = Z_1, \varepsilon \sim \mathcal{N}(0, \sigma_{Z_1}^2), \tag{15}$$

$$X_{2,t-1} = \alpha X_{1,t-2} + \varepsilon_{Z_2}, \varepsilon_{Z_2} \sim \mathcal{N}(0, \sigma_{Z_2}^2),$$

$$X_{3,t} = \beta X_{2,t-1} + \gamma X_{1,t-2} + \varepsilon_{Z_3}, \varepsilon_{Z_3} \sim \mathcal{N}(0, \sigma_{Z_3}^2),$$

where $X_{1,t-2}$ is a confounder affecting both $X_{2,t-1}$ and $X_{3,t}$, and $\alpha, \beta, \gamma$ are constants. Objective: Show how the observational regression coefficient of $X_{3,t}$ on $X_{2,t-1}$ is biased due to $X_{1,t-2}$, and how the intervention $do(X_{2,t-1} = x)$ removes this spurious correlation.

In observational data, the ordinary least squares estimate of $X_{3,t}$ on $X_{2,t-1}$ is:

$$\text{Var}(X_{2,t-1}) = \text{Var}(\alpha X_{1,t-2} + \varepsilon_{Z_2}) = \alpha^2 \sigma_{Z_1}^2 + \sigma_{Z_2}^2, \tag{16}$$

$$\text{Cov}(X_{2,t-1}, X_{3,t}) = \text{Cov}(\alpha X_{1,t-2} + \varepsilon_{Z_2}, \beta(\alpha X_{1,t-2} + \varepsilon_{Z_2}) + \gamma X_{1,t-2})$$

$$+ \text{Cov}(\alpha X_{1,t-2} + \varepsilon_{Z_2}, \varepsilon_{Z_3})$$

$$= \text{Cov}(\alpha X_{1,t-2} + \varepsilon_{Z_2}, (\beta\alpha + \gamma)X_{1,t-2} + \beta\varepsilon_{Z_2})$$

$$= \alpha(\beta\alpha + \gamma)\text{Var}(X_{1,t-2}) + \beta\text{Var}(\varepsilon_{Z_2})$$

$$= \alpha(\beta\alpha + \gamma)\sigma_{Z_1}^2 + \beta\sigma_{Z_2}^2,$$

$$b_{obs} = \frac{\text{Cov}(X_{2,t-1}, X_{3,t})}{\text{Var}(X_{2,t-1})}$$

$$= \frac{\alpha(\beta\alpha + \gamma)\sigma_{Z_1}^2 + \beta\sigma_{Z_2}^2}{\alpha^2\sigma_{Z_1}^2 + \sigma_{Z_2}^2}$$

$$= \beta + \gamma\frac{\alpha\sigma_{Z_1}^2}{\alpha\sigma_{Z_1}^2 + \sigma_{Z_2}^2},$$

where the term $\gamma\frac{\alpha\sigma_{Z_1}^2}{\alpha\sigma_{Z_1}^2 + \sigma_{Z_2}^2}$ is spurious correlation bias due to the confounder $X_{1,t-2}$. Observational regression coefficients depend on the distribution of $X_{1,t-2}$ (on $P(X_{1,t-2} \mid X_{2,t-1})$), and therefore can vary across environments. As $P(X_{1,t-2})$ or the relationship between $X_{2,t-1}$ and $X_{1,t-2}$ changes across environments, this observational slope changes, and OOD predictions fail.

$$X_{3,t} = \beta X_{2,t-1} + \gamma X_{1,t-2} + \varepsilon_{X_3}, X_{2,t-1} \text{ set to } x \text{ exogenously.} \tag{17}$$

The interventional expectation is:

$$\mathbb{E}[X_{3,t} \mid do(X_{2,t-1} = x)] = \mathbb{E}[\beta X_{2,t-1} + \gamma X_{1,t-2} + \varepsilon_{Z_3} \mid do(X_{2,t-1} = x)] \tag{18}$$

$$= \beta x + \gamma\mathbb{E}[X_{1,t-2}] + \mathbb{E}[\varepsilon_{Z_3}]$$

$$= \beta x$$

Thus, the confounder $X_{3,t}$ no longer affects the slope; the spurious correlation term disappears. Hence, interventions remove spurious correlations and reveal the invariant causal effect, which forms the basis for OOD generalization in causal discovery and causal graph-based prediction models.

## A.2 EXPERIMENTS

### A.2.1 DETAILS OF DATASETS

**Synthetic Datasets.** To enable model pre-training, we follow CP (Stein et al., 2024) to construct a synthetic dataset. First, we randomly generate a causal graph G by sampling directed edges between nodes. Then, we assign edge weights by drawing from a specified uniform distribution to obtain the weighted adjacency matrix $Ga$. For each edge, we sample a nonlinear functional relationship $f_{k,j,i}$ from a predefined set of functions:

$$\mathcal{F}^n = \left\{ e^x, \ x^2, \ \sigma(x), \ \sin(x), \ \cos(x), \ \text{relu}(x), \ \log(\sigma(x)), \ \frac{1}{x}, \ \|x\|, \ \text{clamp}(x, (-0.5, 0.5)) \right\}. \tag{19}$$

Table 5: Details of Root Cause Identification Datasets.

| Dataset | Training Time Steps | Test Sequences | Avg. Sequence Length | Avg. # of Root Causes |
|---------|---------------------|----------------|----------------------|-----------------------|
| SWaT (51) | 49,500 | 20 | 51 | 13.35 |
| MSDS (10) | 29,268 | 4,255 | 21 | 3.05 |

For linear relationships, we simply set $f_{k,j,i}(x) = x$. Next, we randomly initialize the time lag for each connection. Using the causal graph, functional mappings, and lag structure, we iteratively generate multivariate time series data. Finally, we synthesize datasets with varying numbers of variables and lags, ensuring diversity in both structural and dynamic properties.

**Real-world Benchmark.** CausalRiver (Stein et al., 2025) treats river flow measurements from monitoring stations as a time series dataset with inherent causal structure. It covers two regions in Germany: the eastern German territory (666 measuring stations) and Bavaria (494 stations). The dataset spans from 2019 to 2023 with a temporal resolution of 15 minutes. Importantly, CausalRiver includes both normal hydrological conditions and extreme events such as heavy rainfall and large-scale precipitation, enabling the study of causal dynamics under diverse environmental scenarios. The dataset is categorized into five types based on the characteristics of their underlying causal structures:

- Random: All connected subgraphs with three or five nodes, covering the entire dataset and full diversity of benchmark conditions.

- Close: Connected subgraphs whose edges have a maximum geographic (Euclidean) distance of 0.3; by excluding long-range connections, causal effects are expected to be more pronounced. This set is fully contained within Random.

- Random + 1: Connected subgraphs with two or four nodes, combined with one additional isolated node. To avoid confounding, the isolated nodes are drawn from coastal or border regions where disconnected nodes naturally occur.

- Root cause: Connected subgraphs with three or five nodes in which each node has at most one parent, forming chain-like structures. This setting is useful for root-cause analysis (Ikram et al., 2022) and is fully contained within Random.

- Confounder: Subgraphs with four or six nodes containing a single node with multiple children (rarely observed in cases such as river splits). The multi-child node is then removed to simulate permanent hidden confounding.

**Root Cause Identification Datasets.** SWaT (Mathur & Tippenhauer, 2016) is a dataset collected from a testbed that simulates a real-world water treatment plant. It comprises data from 51 sensors in the critical infrastructure system during continuous operation, including both normal operating conditions and attack scenarios within the water treatment process. MSDS (Multi-Source Distributed System) (Nedelkoski et al., 2020) is developed on an OpenStack testbed and serves as a dataset for AIOps (Artificial Intelligence for IT Operations). Instances of fault injections in this system are labeled as anomalies. More detail are provided in Table 5.

### A.2.2 MORE DETAILS OF METRICS ON ROOT CAUSE IDENTIFICATION

Given a multivariate time series $\mathcal{X}$, the $Recall@K$ is defined as:

$$Recall@K = \frac{1}{|\mathcal{X}|} \sum_{x_i \in \mathcal{X}} \frac{\left| V_{x_i}^{(RC)} \cap \{R_{x_i}[k] \mid k = 1, 2, \ldots K\} \right|}{\min \left( K, \left| V_{x_i}^{(RC)} \right| \right)}, \quad (20)$$

where $R_{x_i}[k]$ indicates the time series at the $k$-th rank for the channel $x_i$, and $V_{x_i}^{(RC)}$ indicates a set of root cause variables over the whole channel $x_i$. Note that if a time series receives multiple exogenous interventions, it only counts as one root cause time series in $V_{x_i}^{(RC)}$. We further compute the overall performance by computing the average $Recall@K$, denoted as $Avg@K = \frac{1}{K} \sum_{k=1}^{K} Recall@k$.

Table 6: Ablation studies on Root Cause Localization. w/o inter, w/o intra, w/o exogenous, w/o causal mixup, w/o intervention represent removing the inter-window attention, intra-window attention, exogenous, causal mixup, and intervention, respectively.

| Dataset | Model | Recall@1 | Recall@3 | Recall@5 | Recall@10 | Avg@10 |
|---------|-------|----------|----------|----------|-----------|--------|
| MSDS | w/o inter | 0.452 | 0.792 | 0.910 | **1.000** | 0.492 |
| | w/o intra | 0.498 | 0.801 | 0.910 | **1.000** | 0.835 |
| | w/o exogenous | 0.227 | 0.697 | 0.893 | **1.000** | 0.694 |
| | w/o causal mixup | 0.500 | 0.792 | 0.993 | **1.000** | 0.842 |
| | w/o intervention | 0.296 | 0.798 | 0.900 | **1.000** | 0.728 |
| | T-Caus | **0.515** | **0.993** | **0.996** | **1.000** | **0.929** |
| SWaT | w/o inter | 0.200 | 0.350 | 0.350 | 0.455 | 0.351 |
| | w/o intra | 0.250 | 0.375 | 0.375 | 0.450 | 0.360 |
| | w/o exogenous | 0.100 | 0.150 | 0.250 | 0.350 | 0.205 |
| | w/o causal mixup | 0.250 | 0.330 | 0.450 | 0.455 | 0.342 |
| | w/o intervention | 0.150 | 0.220 | 0.330 | 0.375 | 0.252 |
| | T-Caus | **0.300** | **0.450** | **0.450** | **0.475** | **0.440** |

### A.2.3 IMPLEMENTATION DETAILS

We first pretrain T-Caus using the Adam optimizer (Kingma & Ba, 2015) with a learning rate of $10^{-3}$. We apply early stopping with a patience of 30 epochs to prevent overfitting. The loss weights $\lambda_G$, $\lambda_{KL}$ and $\lambda_{do}$ are set to 0.5. The model consists of 4 alternating attention blocks, and the number of Gaussian components $K$ is set to 10. For fine-tuning in the causal discovery task, we use a learning rate of $10^{-4}$, and keep $\lambda_G$ and $\lambda_{KL}$ fixed at 0.5. In the root cause identification task, fine-tuning is also performed with a learning rate of $10^{-4}$, and $\lambda_{KL}$ is set to 0.5. All experiments are implemented in PyTorch and conducted on an NVIDIA A800 80GB GPU. Following AERCA (Han et al., 2025), data preprocessing is standardized across datasets using a MinMax scaler. To improve computational efficiency, we downsample the SWaT dataset every 10 seconds and the MSDS dataset every 5 time steps.

### A.2.4 MORE ABLATION STUDY

To investigate the impact of different components on root cause identification, we conduct an ablation study focusing on inter-window attention, intra-window attention, mixture-of-Gaussians modeling of exogenous variables, causal mixup, and intervention. As shown in Table 6, we find that the intervention mechanism achieves superior performance in identifying root causes. This gain is attributed to the fact that interventions provide the model with information across diverse environments, enabling it to filter out spurious correlations, learn more accurate causal relationships, and enhance overall robustness. Furthermore, when the mixture-of-Gaussians constraint on exogenous variables is removed, T-Caus exhibits a notable drop in performance, demonstrating the importance of instance-specific exogenous variable estimation for effective root cause identification.

