# OpenReview forum: "Time Series Causal Discovery via Instance-specific Modeling and Intervention-based Pretraining"
_ICLR.cc/2026/Conference — ICLR 2026 Conference Withdrawn Submission_

### Official Review · Reviewer_Rnph · 2025-10-30

**Soundness:** 2
**Presentation:** 3
**Contribution:** 3
**Rating:** 4
**Confidence:** 4

**Summary:**

This paper proposes T-Caus, a framework for temporal causal discovery that aims to address two key challenges: the inherent complexity of temporal causal dependencies and generalization under distribution shifts. To tackle these, the authors introduce a hierarchical model with dual-scale representation learning and instance-specific exogenous variable estimation, complemented by a pre-training strategy using causal intervention and mixup for improved generalization.

**Strengths:**

Strengths:

- The paper identifies two important challenges in time series causal discovery in capturing complex, multi-scale dependencies and achieving causality-aware generalization, which aim to be addressed in the proposed framework.

- The proposed solutions, including the instance-specific exogenous variable estimation and the causal mixup for data augmentation, show some promising results in the experiments.

**Weaknesses:**

Weaknesses:

- The contribution of the paper is somewhat unclear. While it states the two challenges, it is also disclosed in many works, e.g., [1-2], making it hard to situate T-Caus's contributions within the current literature.

- The connection between the proposed methods and the proposed solutions is not clear. For instance, why the intervention-based pretext task can address the distribution shift challenge, which is a lack of theoretical proof.

- The paper presents some theoretical claims, such as Theorem 1. However, the claim of the intervention is well-known, and it also did not explain why the intervention is needed, because using the back-door criterion can also eliminate the spurious associations using pure observational data.

- Moreover, the proof of Theorem 1 only discusses the linear cases, which lack mathematical rigor for the general nonlinear scenario.

- The empirical evaluation is performed on a few real-world datasets, making it hard to assess the method's generalizability. As suggested, the experiments would be significantly strengthened by including comprehensive tests on synthetic data. This would allow for a controlled analysis of performance on specific causal structures, such as disentangling accuracy on inter-window versus intra-window causal links.

[1] D'Acunto, Gabriele, et al. "Learning Multiscale Non-stationary Causal Structures." TRANSACTIONS ON MACHINE LEARNING RESEARCH (2023).

[2] Huang, Biwei, et al. "Causal discovery from heterogeneous/nonstationary data." Journal of Machine Learning Research 21.89 (2020): 1-53.

**Questions:**

See the weaknesses above.

---

### Official Review · Reviewer_gGBj · 2025-10-31

**Soundness:** 1
**Presentation:** 1
**Contribution:** 2
**Rating:** 2
**Confidence:** 5

**Summary:**

The paper introduces T-Caus, a pre-training-based approach to temporal causal discovery. The key innovations include (i) a dual-scale attention block, (ii) an instance-specific MoG latent for “exogenous noise,” and (iii) a pretraining stage with a synthetic “intervention” pretext task plus causal mixup. The authors report results on one causal discovery dataset, and one for root cause analysis.

**Strengths:**

The authors employ an interesting architecture to unify temporal causal modeling, noise modeling, and OOD-training. Some of the ablation studies are interesting in understanding the contribution of the different parts of the architecture.

**Weaknesses:**

1. There are **several** incorrect claims and assertions throughout the paper.

- The claim that existing models do not handle cascading temporal effects is *wrong* and needs more justification to clarify why the authors make this assertion. Methods like PCMCI already condition on all variables across lags up to a max lag, so by design, they test edges while controlling for multi-step pathways. If the authors think this modeling is *insufficient* empirically, then they should *explicitly show this* with experiments that demonstrate this behavior.
Similarly, "Existing approaches (Han et al., 2025), which typically assume a fixed Gaussian prior, lack the
flexibility to adapt to such diversity and cannot model exogenous factors across multiple series." is misleading since it ignores many different works (for example [1]) that explicitly model the exogenous factors.
- The entire paragraph in the introduction related to "Challenge 2" is full of statements that do not follow from one another. First of all, why do the authors bring up Vicinal Risk Minimization (VRM) to justify the perceived poor OOD performance of these models? This is a moot question for many causal discovery algorithms that are *completely unsupervised*, hence the question of OOD doesn't even apply to them. The authors then say "However, existing time series causal discovery methods fail to account for both the diversity of individual series and the variability of causal relationships across different series." This statement has *nothing* to do with the previous statement about VRM. Furthermore, this statement is disingenuous because it ignores the *variety* of existing work [2, 3, 4, 5, 6] that handle various degrees of variation in the individual series as well as variation across different series.
The authors also state: "Unfortunately, existing deep learning methods lack effective constraints to identify and differentiate spurious correlation from causation." This is, once again, a poorly supported statement that ignores the **fundamental aim** of most causal discovery algorithms (i.e. distinguishing correlation from causation).
- In the related work section, the authors claim: "Granger causality-based approaches share a fundamental limitation: they do not explicitly model endogenous errors or exogenous noise". Once again, this is **completely inaccurate**. In fact, one of the papers that the authors extensively cite [7] is a Granger causal model that explicitly models the exogeneous noise.

Overall, the entire introduction needs to be rewritten to remove these unsupported claims.


2. A lot of related work is missed out. In addition to the papers mentioned previously [1, 2, 3, 4, 5, 6, 7], the authors ignore the line of work on causal discovery from a mixture of DAGs [8, 9] which partially model the "variability of causal relationships across different series" (samples) albeit in the IID case instead of the time-series case. Most egregiously, the authors ignore the line of work on amortized causal graph inference [10, 11] which deal with a very similar setting (pretraining for causal discovery).

3. The problem setting is unclear. The paper emphasizes multi-scale dependencies and evolving relations, but the method appears to output a single graph per instance. There is no explicit mechanism for regime switches or mixtures of graphs across windows/environments, despite the claims around non-stationarity.

4. The time-series mix-up idea (Lines 310-322) is not principled. Mixing examples does not necessarily preserve the causal structure unless the SCM has linear relations.

5. The theory result (Theorem 1) The statement "intervention blocks spurious path" is a direct restatement of the standard backdoor argument and not novel. Worse, the proof explicitly assumes a linear SCM in Appendix A.1, so it doesn’t support the general claims made in the main text.

6. The causal discovery results rely on fine-tuning: "Our model is fine-tuned on the Bavaria dataset and evaluated on the Eastern Germany river datasets". This seems unfair to other models, which perform causal discovery in an unsupervised fashion on the target dataset. Moreover, in most practical applications, the ground-truth causal graph is unavailable, making this setup quite unrealistic. The authors should report the 0 shot performance of their model (after synthetic pretraining) to make it fair to the baselines.

### References
[1] Gong, Wenbo, et al. "Rhino: Deep causal temporal relationship learning with history-dependent noise." arXiv preprint arXiv:2210.14706 (2022).

[2] Mameche et al., SPACETIME: Causal Discovery from Non-Stationary Time Series, AAAI 2025.

[3] Varambally, Ma, Yu, Discovering mixtures of structural causal models from time series data, arXiv:2310.06312, 2023.

[4] Huang et al., Causal discovery from heterogeneous/nonstationary data, JMLR 2020.

[5] Gao et al., Causal discovery in semi-stationary time series, NeurIPS 2023.

[6] Saggioro et al., Regime-dependent causal relationships from observational time series, Chaos 2020.

[7] Han, Xiao, et al. "Root Cause Analysis of Anomalies in Multivariate Time Series through Granger Causal Discovery." The Thirteenth International Conference on Learning Representations. 2025.

[8] Saeed et al., Causal structure discovery from distributions arising from mixtures of DAGs, ICML 2020.

[9] Strobl, Causal discovery with a mixture of DAGs, ML 2023.

[10] Ke et al., Learning to Induce Causal Structure, arXiv 2022.

[11] Lorch et al., Amortized Inference for Causal Structure Learning, NeurIPS 2022.

**Questions:**

Apart from the major weaknesses pointed out, I have the following comments/questions:

1. The notation $s_t$ and $\tilde{s}_t$ seem to be overloaded and used for both the reconstruction and intermediate representations.
2. "The means μ = 0 and variances Σ of the K Gaussian component is treated as learnable parameters:" Is the mean enforced to be 0 or learned?

---

### Official Review · Reviewer_USYY · 2025-11-01

**Soundness:** 2
**Presentation:** 2
**Contribution:** 2
**Rating:** 4
**Confidence:** 3

**Summary:**

The paper presents T-Caus, a pretraining framework for time series causal discovery based on structural causal models (SCMs), emphasizing hierarchical instance-specific modeling with dual-scale attention for temporal dependencies and Gaussian mixtures for exogenous noise, alongside intervention-based pretraining and causal mixup for generalization under distribution shifts. While aiming to address challenges in capturing cascading causal relations and heterogeneity, the methodological contributions suffer from underspecified formulations and vague terminology, rendering key claims unverifiable without clearer definitions.

**Strengths:**

Incorporates diverse concepts such as inter- and intra correlations, intervention-aware causal discovery mechanism, causal mixup mechanisms.

Performed empirical studies based on several datasets to show better performance.

**Weaknesses:**

This paper is difficult to read. While I appreciate the authors' efforts to incorporate as many techniques as possible to address nearly all limitations of existing approaches, it lacks a coherent framework to justify the proposed method.

The core 'instance-specific' modeling lacks a precise definition; it is vaguely described as routing mixing coefficients from enhanced representations in a Gaussian mixture (Eq. 7), without clarifying how this differs from standard conditional priors or ensuring the identifiability of exogenous distributions. This leads to potential confusion with global mixtures.


The SCM formulation (Eq. 1) remains abstract, with nonlinear functions  and the mechanics of interventions on generative processes (Eq. 6) left undefined in form or sampling. This makes it unclear how the model enforces causal invariance or avoids circular dependencies in reconstruction (Eq. 8).

Descriptions of generalization components, such as causal invariance in interventions, are superficial and rely on unproven assumptions about environmental stability, without deriving error bounds or discussing identifiability under hidden confounders. Meanwhile, hyperparameters like K=10 and loss weights (Eq. 12) appear ad hoc.

**Questions:**

None.

---

### Official Review · Reviewer_dYW4 · 2025-11-02

**Soundness:** 2
**Presentation:** 2
**Contribution:** 2
**Rating:** 4
**Confidence:** 3

**Summary:**

The authors developed a hierarchical framework for time series causal discovery based on Structural Causal Models, integrating a dual-scale iterative attention mechanism and a Gaussian mixture model to capture complex temporal causal dependencies and address heterogeneous exogenous distributions. Experimental results demonstrate the superior performance of T-Caus in both causal discovery and root cause identification tasks.

**Strengths:**

1. The paper proposes a pretraining framework, which integrates various advanced techniques (such as dual-scale iterative attention mechanism, Gaussian mixture model, and causal mixture strategy) to address complex challenges in causal discovery for time series.
2. The authors validated T-Caus on multiple out-of-distribution datasets, showcasing its superior performance in causal discovery and root cause identification tasks.

**Weaknesses:**

1. The presentation of the paper lacks clarity, and it is not clear what specific problem the paper aims to address. While the title focuses on causal discovery, the introduction and related work sections also discuss causal effect estimation. It is important to note that causal discovery and causal effect estimation are two distinct problems. The authors are advised to clearly distinguish between the two and focus on the core task of the paper.
2. The paper does not provide a theoretical proof or analysis to demonstrate under what conditions the T-Caus framework can uniquely identify the true causal structure in time series, rather than merely finding pseudo-causal relationships that fit the observed data.
3. The paper lacks a comprehensive review of related literature. For example, the authors claim that "existing time series causal discovery methods fail to account for both the diversity of individual series and the variability of causal relationships across different series." However, the paper "Specific and Shared Causal Relation Modeling and Mechanism-Based Clustering" specifically addresses this issue. It is recommended that the authors include this relevant work in their literature review to ensure comprehensive coverage and clarify the distinction and contribution of their approach.
4. The experimental section lacks important details. Specifically, the selection of baseline methods is not comprehensive, as it does not include comparisons with some of the latest causal discovery methods, making it difficult to fully demonstrate the advantages of the proposed approach. Additionally, the paper does not provide a sensitivity analysis of the hyperparameter settings, which is necessary to validate the reasonableness and robustness of the chosen parameters.

**Questions:**

See the weaknesses above.

---

### Note · Authors · 2026-01-20

**Comment:**

We sincerely thank the reviewers and the program committee for their constructive feedback and would like to withdraw the submission at this stage to conduct further revisions and additional experiments.

**Withdrawal Confirmation:**

I have read and agree with the venue's withdrawal policy on behalf of myself and my co-authors.